# Allometric Growth Pattern and Hunger Tolerance of *Hemibarbus maculatus* Bleeker Larvae

**DOI:** 10.3390/biology13030164

**Published:** 2024-03-03

**Authors:** Min Xie, Pengpeng Wang, Qi Deng, Si Liu, Zhou Zhang, Hao Wu, Jing Xiang, Jie Zhou, Xin Yang, Rui Song, Shaoming Li, Zhonggui Xie

**Affiliations:** 1Hunan Fisheries Science Institute, Changsha 410153, China; xieminhaha@126.com (M.X.); 111_bnm@163.com (P.W.); superdq1990@126.com (Q.D.); 18273162664@163.com (S.L.); zz19961022@hotmail.com (Z.Z.); wh17380133463@163.com (H.W.); xiangjin1023@163.com (J.X.); yangxinhzau@163.com (X.Y.); lishaoming1977@126.com (S.L.); xieice123@163.com (Z.X.); 2Hunan Aquatic Foundation Seed Farm, Changsha 410153, China; 3College of Animal Science and Technology, Hunan Agricultural University, Changsha 410125, China; 17674042583@163.com

**Keywords:** allometric growth, *Hemibarbus maculatus*, infection point, point of no return, primary feeding rate

## Abstract

**Simple Summary:**

To clarify the allometric growth pattern and hunger tolerance of *Hemibarbus maculatus* Bleeker larvae, the morphological lengths of their functional organs were measured continuously and their primary feeding rates under a state of starvation were studied. A control group and starvation group were set up for this study, and 10 larvae were sampled from each group every day in order to study their allometric growth pattern and hunger tolerance. The results showed that the *Hemibarbus maculatus* larvae opened their mouths for feeding at 4 DAH (days after hatching), and the yolk sac disappeared completely at 11 DAH. The PNR (point of no return) of the *Hemibarbus maculatus* larvae was 12–13 DAH, and the ratio of the DAH in the mixed trophic period compared to the endotrophic period was 1.75, indicating that *Hemibarbus maculatus* larvae have strong starvation tolerance. *Hemibarbus maculatus* larvae preferentially developed their heads, fins, and eyes, related to the functions of feeding, balancing, and swimming, in order to cope with complex environments. Therefore, when considering a water temperature of 22.66 ± 1.56 °C, 4–5 DAH is the best time to cultivate in the pond, which should not be carried out later than 12 DAH.

**Abstract:**

To clarify the allometric growth pattern and hunger tolerance of *Hemibarbus maculatus* Bleeker larvae, the morphological lengths of their functional organs were measured continuously and their primary feeding rates under a state of starvation were studied. A control group and starvation group were set up for this study, and 10 larvae were sampled from each group every day in order to study their allometric growth pattern and starvation tolerance. The results indicated that the *Hemibarbus maculatus* larvae opened their mouths for feeding at 4 days after hatching, and that the yolk sac disappeared completely at 11 days after hatching. The *Hemibarbus maculatus* larvae preferentially developed their heads, fins, and eyes, related to the functions of feeding, balancing, and swimming, in order to cope with complex environments. The growth inflection points for the head length, pectoral fin length, dorsal fin length, eye diameter, eye spacing, snout length, and body height were characterized by total lengths of 10.93 mm, 11.67 mm, 11.67 mm, 13.17 mm, 16.53 mm, 15.13 mm, and 15.13 mm, respectively. Prior to and following the inflection point, positive allometric growth was observed in all organs. After the inflection point, the dorsal fin continued to maintain positive allometric growth, while the others changed to isometric allometric growth. A growth inflection point was not observed for trunk length or the lengths of the tail and anal fins. The trunk length always maintained negative allometry, while the tail and anal fin lengths were reversed. The growth inflection point of the tail length was at a total length of 13.68 mm. Before and after the growth inflection point, negative and isometric allometric growths were observed, respectively. According to the relationship between the total length and number of days after hatching, the growth inflection point of the *Hemibarbus maculatus* larvae was concentrated at TL = 10.93–16.53 mm, which was observed 14–20 days after hatching. The point of no return for the *Hemibarbus maculatus* larvae was 12–13 days after hatching, and the ratio of days after hatching in the mixed trophic period to the endotrophic period was 1.75, indicating that the larvae had strong hunger tolerance. Therefore, when considering a water temperature of 22.66 ± 1.56 °C, 4–5 days after hatching is the best time to cultivate in the pond, and it should not be carried out later than 12 days after hatching.

## 1. Introduction

Food and the early development environment are the key factors that determine the early survival rate of fish [1,2]. Allometric growth is a phenomenon in which different organs grow at different rates [3]. In the early development of fish, allometric growth usually refers to the growth rate of an organ relative to its full length [4,5,6,7]. In particular, allometric growth reflects the strategy through which larvae preferentially develop certain organs in order to adapt to complex living environments in their early life stages [8,9,10]. It is generally believed that the organs that develop first in the early stages of larval growth are those that are important for survival. For example, to ensure the development of respiration and feeding organs, *Brycon orbignyanus*, *Misgurnus anguillicaudatus*, and *Siniperca chuatsi* preferentially develop their heads [2,11,12]; meanwhile, *Oplegnathus fasciatus* and *Lates calcarifer* prioritize the development of visual organs to improve their predation and avoidance abilities [13,14]. Most fish preferentially develop their heads and tail lengths while slowing down trunk length growth to control their body length and improve their locomotor ability [2,15,16]. The point of no return (PNR) reflects the ability of early larvae to tolerate hunger: the larger the PNR, the stronger the ability of early larvae to tolerate hunger [17,18]. The PNR of larvae is related to the fish species and water temperature. For example, the PNR of *Hucho taimen* is 39–40 days old at a water temperature of 10.0–11.0 °C, while the PNR of Pacific cod is 9 days old at the same water temperature. [19,20]. When the water temperature is 18.5–21.9 °C, the PNR of *Percocypris pingi* is 17 days old [17], and the PNR of the larvae of *Acanthopagrus schlegelii* is 7 days old when the water temperature is 18–19 °C [21].

*Hemibarbus maculatus* Bleeker, a member of the family Cyprinidae and *Hemibarbus*, is an important small economic fish in China, which is mainly distributed in rivers, lakes, and reservoirs in the plain area of the middle and lower reaches of the Yangtze River [22]. Due to its fresh and tender meat and delicious taste, *Hemibarbus maculatus* Bleeker has become a popular fish species in aquaculture [23]. Before the implementation of the “10-year fishing ban” policy, due to over-fishing and ecological environment deterioration, the habitats and spawning grounds of *Hemibarbus maculatus* were destroyed. The population resources of *Hemibarbus maculatus* showed a declining trend, and its fishery resources appeared to show minimizing and younger age trends [24]. Li et al. also stated that *Hemibarbus maculatus* was in a state of over-exploitation [24]. Obtaining information on the early ontogeny of fishes is important to better understand the dynamics of fish populations [2,25]. Early studies on *Hemibarbus maculatus* Bleeker have mainly focused on its growth, resource quantities, biological characteristics, and artificial breeding [24,26]. However, there have been no reports on their early development pattern and ability to tolerate hunger, which are important for the cultivation and resource conservation of *Hemibarbus maculatus* Bleeker. Obtaining knowledge on allometric growth patterns during the early life history stages of a species provides essential information regarding the adaptation strategies of fish populations [2]. Therefore, studying the allometric growth pattern and PNR is helpful to understand the early development pattern and internal strategies through which *Hemibarbus maculatus* Bleeker copes with complex and volatile living environments, which has great significance for improving the survival rates of its larvae.

## 2. Materials and Methods

### 2.1. Fish Rearing and Sampling

The experiment was conducted from 25 April to 26 May 2020 at the breeding base of Hunan Fishery Science Institute in Changsha, China. The larvae were incubated in a breeding base. The eggs were incubated in water with a temperature ranging from 18.3 to 20.0 °C. The healthy larvae were randomly divided into two groups (control group and starvation group) and placed in six fiberglass aquariums (55 cm × 44 cm × 55 cm). There were three replicates in each group and 1000 larvae were released in each replicate. After the experiment began, one-third of the water was replaced every 24 h. The larvae were recorded as 1 day old at 24 h after hatching. The control group was fed excessive artificial cultivated plankton every day from 1–18 days after hatching (DAH), after which they were fed commercial powder (Produced by Zhejiang Jinjia Aquatic Feed Co., Ltd., Hangzhou, China. Crude protein *≥* 43.0%, crude lipid *≥* 5.0%, moisture *≤* 10.0%, Ash *≤* 18.0%) twice a day (at 9:00 and 17:30). The larvae in the starvation group were fasted until all had died. During the experiment, the water temperature was 22.66 ± 1.56 °C, the pH was 7.3–7.8, and the dissolved oxygen was ≥5 mg/L.

In the control group, 10 larvae were sampled at 9:00 every day from 1–32 DAH. In the starvation group, 10 larvae from each replicate were placed in 500 mL beakers per day from 1 DAH. Artificially cultured plankton were placed into the beaker for overfeeding, and the larvae were collected 1 h later. The sampled larvae from both groups were directly fixed in 4% paraformaldehyde for further analysis [17,27].

### 2.2. Measurements

To reduce variation in shrinkage rate among specimens, the sampled larvae and juveniles were observed and measured after being preserved for 10 days. All samples were photographed under a stereomicroscope (Olympus SZ61) and measured using Image-Pro Plus software 6.0. The morphological indices measured included total length (TL), head length (HL), snout length (SL), eye diameter (ED), eye distance (EDS), body height (BH), trunk length (TrL), pectoral fin length (PL), dorsal fin length (DL), caudal fin length (CL), anal fin length (AL), and tail length (TaL), and were measured according to the method of Xu et al. [28]. All measurement lengths were accurate to 0.01 mm (Figure 1). The larvae in the starvation group were dissected under the stereomicroscope (Olympus SZ61) in order to observe whether there was food in their intestine. The primary feeding rate (PFR) was calculated as follows:PFR = n/N × 100 × 100%,
where n refers to the number of feeding larvae, and N refers to the total number of larvae. PNR was determined according to the method of Zhang et al. [17].

### 2.3. Data Analysis

SPSS Statistics 23.0 software was used to analyze the relationship between DAH and the total length of the larvae in the control group, and the function with the largest R^2^ value was used as the growth equation [29]. Allometric growth patterns were described using the growth coefficient, which was calculated as a power function Y = aX^b^, where X denotes total length, Y denotes the morphological indices, a is the intercept, and b is the allometry index [2,30]. Isometric allometric growth is inferred when b = 1, positive allometric growth when b > 1, and negative allometric growth when b < 1. All data conformed to the normal distribution when tested using the Origin 2021 software. Before the t-test, the homogeneity of variance was assessed; when the Sig value was greater than 0.05, the variance could be considered homogeneous. Then, a t-test was conducted to determine whether there was significant difference between the growth index (b) and 1. The growth inflection point was determined according to the methods of Van Snik et al. and Shea and Vecchione [9,31].

## 3. Results

### 3.1. Relationship between TL and DAH

Curve model fitting and parameter estimation of the relationship between TL and DAH for the *Hemibarbus maculatus* larvae were performed using “Curve Estimation” in SPSS Statistics 23.0 software. The best function was Y = 5.61 × 0.54^X^ R^2^ = 0.98. The total length of the newly hatched *Hemibarbus maculatus* larvae was 6.24 ± 0.14 mm. After 32 days of growth, the total length reached 30.02 ± 2.82 mm (see Figure 2).

### 3.2. Allometric growth

A total of eight of the eleven morphometric characteristics that were measured presented inflection points during the study period (Figure 3). The growth inflection point of the *Hemibarbus maculatus* larvae was concentrated at TL = 10.93–16.53 mm. Furthermore, seven of the eight morphometric indicators with inflection points followed a positive allometric growth trend before the inflection point, with the tail length (TaL) being the exception (Figure 3).

Head organs: An inflection point was observed for the allometric growth function of each measured head organ. The inflection points of ED, SL, and EDS were observed at 13.17 mm, 15.13 mm, and 16.35 mm TL, respectively (Figure 3a–c). Before the inflection points, the allometry indices b_1_ were 1.50, 1.75, and 1.79, which were significantly different from 1 (*p* < 0.05) and demonstrated positive allometry. After the inflection points, the allometry indices b_2_ were 0.97, 0.94, and 0.74, respectively. There was no significant difference between the allometry indices of SL and ED and 1 (*p* > 0.05), indicating isometric allometric growth; however, EDS showed negative allometry (*p* < 0.05).

Part of body: A growth inflection point was not observed for TrL, which showed negative allometry (b = 0.76). The inflection points of HL, BH, and TaL were observed at 10.93 mm, 15.13 mm, and 13.68 mm, respectively (Figure 3d–g). HL and BH showed positive allometric growth at first (b_1_ = 1.85 and 1.74, respectively), then changed to isometric allometric growth (b_2_ = 0.98 and 1.10, respectively). The b_1_ and b_2_ values for TaL were 0.65 and 1.14, which means that TaL first showed negative and then positive allometry.

Fins: There were no growth inflection points for CL and AL. Their allometry indices were 1.62 and 2.21, respectively, indicating positive allometric growth. The growth inflection points for PL and DL were 11.67 mm and 16.35 mm TL. DL always showed positive allometric growth, both before and after the inflection point (b_1_ = 3.08, b_2_ = 1.31). After the inflection point, PL changed from positive to isometric allometry (b_1_ = 1.36, b_2_ = 1.06) (Figure 3h–k).

### 3.3. Effects of Starvation on Feeding and Growth of Larvae of Hemibarbus maculatus

Considering a water temperature of 22.66 ± 1.56 °C, the PFR of *Hemibarbus maculatus* larvae in the starvation group is shown in Figure 4. The PFR was 10% at 4 DAH. Subsequently, the PFR gradually increased to the highest level at 11 DAH (90%), then rapidly decreased to half of the highest level (45%) at 12–13 DAH. Therefore, the PNR was between 12–13 DAH. At 0–3 DAH, the larvae of *Hemibarbus maculatus* were completely nourished by their yolk sac. Therefore, the TL of larvae in the starvation and control groups showed only innate individual differences. From 4–10 DAH, a mixed nutrition stage was observed. The larvae in the control group grew rapidly, while those in the starvation group grew slowly, depending only on the yolk sac. After 10 DAH, with the disappearance of the yolk sac, larvae entered the complete ectotrophic stage. The growth rate of larvae in the two groups was polarized, and the TL of the control group larvae was significantly higher than that of those in the starvation group (Figure 5).

## 4. Discussion

Larvae in the wild face numerous challenges for survival, including food scarcity and predation. To overcome these external factors and ensure population reproduction, fish have developed a range of adaptive evolutionary characteristics throughout their long evolutionary history [32,33]. Allometry serves as an intrinsic developmental strategy in early fish development, helping fish to quickly adapt to the environment. Information on the early ontogeny of fishes is essential to better understand the recruitment success and dynamics of fish populations [25]. The allometric growth pattern of a fish is closely related to its early life characteristics [2]. The development of organs that aid in evading predators and enhance survival rates is prioritized, thus ensuring the survival of early larvae [2,29,34,35]. For example, pelagic larvae usually experience rapid development in their anterior and posterior regions to generate an elongated body, improving their swimming capabilities and feeding success [9,25]. Therefore, knowledge of the allometric growth patterns of fish is helpful in understanding the adaptation strategies of fish populations.

In fish, the development of the head is closely related to vision, feeding, the nervous system, and information processing functions [36]. It serves as the foundation for the development of feeding and respiratory organs [8]. Among all measured organs, *Hemibarbus maculatus* showed an inflection point in head growth at 10.93 mm, which was the first to develop. Previous studies have also shown that many species of fish exhibit preferential head development [6,37,38]. The rapid growth of head length provides ample space for important organs such as the mouth, gills, and brain, ensuring the establishment of respiration and feeding abilities in larvae [37]. Similar results have been confirmed in studies on *Cyprinus carpio* [4], *Clarias gariepinus* [8], and Furong crucian carp (*Carassius auratus* Furong carp ♀ × *Cyprinus carpio* red crucian carp ♂) [36]. As a predation organ, the mouth of fish is very important for feeding [39]. Generally, snout length is used to reflect mouth development in larvae [36,38,40]. In this study, the inflection point of snout length for the larvae was 15.13 mm. Prior to and following this inflection point, positive allometric growth and isokinetic growth were observed, respectively. This suggests that the initial development of the mouth occurred when the total length reached 15.13 mm. The degree of eye development directly influences a larvae’s efficiency with respect to hunting and evading predators. Eye diameter and distance are commonly employed indicators for assessing eye development [4,36]. In our experiment, we identified inflection points for eye diameter at 13.17 mm and eye distance at 16.53 mm. Prior to these inflection points, both eye diameter and distance exhibited positive allometry trends; however, after reaching these points, eye diameter transitioned into isokinetic growth, while eye distance shifted towards a negative allometry pattern. These findings suggest that larval eyes undergo preliminary development when the total length reaches approximately 16.53 mm, consistent with observations in Siberian sturgeon and *Miichthys miiuy* [7,37]. The development of the body parts of *Hemibarbus maculatus* larvae roughly conformed to the “U” growth pattern; that is, while the trunk length grows slowly, the head length and tail length grow rapidly. However, the tail length of *Hemibarbus maculatus* larvae showed negative allometric growth at first, then changed to isokinetic growth. It may be that swimming ability is not particularly important for larvae of *Hemibarbus maculatus*. Similar results have been reported in previous studies on *Carassius auratus* Furong crucian carp [36], *Siniperca chuatsi* [2], *Sebastes schlegelii* [16], and *Percocypris pingi* [15]. The slow growth of the trunk could shorten the full length of larvae and make the movement of larvae more coordinated, which is of great significance in terms of improving their ability to hunt and avoid enemies [30]. The rapid growth of body height was conducive to providing sufficient space for the development and expansion of the swim bladder and the digestive system, thus promoting the movement and digestive abilities of the larvae [33]. Shan et al. [37] have also pointed out that the division of digestive tract structure and function, as well as intestinal curling and folding, could promote the rapid growth of body height. In this study, the body height presented positive allometry at first, then changed to isometric growth after the inflection point (TL = 15.13 mm). This indicates that the swim bladder and digestive organs of *Hemibarbus maculatus* larvae were fully developed when the total length was 15.13 mm, consistent with results observed in *Percocypris pingi* [13], *Sebastes schlegelii* [16], and *Siniperca chuatsi* [2]. The fins of most fish grow in a positively allometric manner before the growth inflection point, then present positive allometric or isometric growth after the inflection point, or positive allometric growth with a slightly lower growth coefficient [28]. As observed in our study, the caudal and anal fin lengths always maintained positive allometric growth, while the pectoral fin first presented positive allometric growth, then changed to isokinetic growth. The growth coefficient of dorsal fin length after the inflection point is generally smaller than that before the inflection point. Of course, there are some exceptions; for example, the pectoral fins of Furong crucian carp and *Lates calarifer* always present isokinetic growth [14,36]; the caudal fin of the spotted knifejaw changes to negative allometric growth after the inflection point [8]; and the caudal fin length of the golden pompano presents negative allometric growth before the inflection point [41]. Fins are organs that larvae use to swim and keep balance [25,42]. The allometric growth model of the fins is related to the change in swimming pattern, environmental pressure, and the size of the fish [25,28,43]. The development of *Hemibarbus maculatus* fins improves its hunting and escaping abilities, thereby increasing its survival rate.

Starvation is the chief factor leading to the death of larvae in the early development stages of fish [17,19,44]. Under the conditions of this experiment, *Hemibarbus maculatus* larvae at 0–3 DAH were in the endogenous stage, while 4–10 DAH was the mixed trophic period and, at 11 DAH, the yolk sac disappeared and they entered the exogenous stage of nutrition. There was no obvious difference in the total length between control and starvation groups between 0–5 DAH, which may be due to the fact that the yolk sac mainly supplied nutrition at this stage. This was similar to the results reported in Furong crucian carp [36]. With the reduction of yolk sac, the total length of *Hemibarbus maculatus* larvae in the two groups showed a clear difference, suggesting that starvation could affect the total length of *Hemibarbus maculatus* larvae. This is consistent with the results of starvation experiments in *Percus cyprinus* [19] and *Plectropomus leopardus* [29]. In order to more objectively reflect the hunger tolerance of different species of larvae, Shan et al. have proposed that the ratio (R) of DAH in the mixed trophic period to the endotrophic period should be used as the criterion for discrimination [45]. If R > 1, it indicates that larvae are less susceptible to starvation stress; otherwise, they are vulnerable to starvation stress. In this study, R = 1.75 > 1, indicating that the *Hemibarbus maculatus* larvae were less susceptible to starvation stress [45]. PNR is an important index to measure the ability to tolerate hunger [46,47]. In this study, the first feeding rate of *Hemibarbus maculatus* larvae was 10% at 4 DAH, while the highest was 90% at 11 DAH, followed by a rapid decline to 20% at 13 DAH. Therefore, the PNR was located between 12–13 DAH. The PNR of *Hemibarbus maculatus* larvae was similar to that of *Squaliobarbus curriculus* [48], *Cichlasoma managuense* [46], and Furong crucian carp [36]; higher than *Oplegnathus fasciatus* [49], *Trachinotus ovatus* [50], and *Plectropomus leopardus* [41]; and lower than *Onychostoma sima* [51] and *Monopterus albus* [52]. The obtained results indicate that *Hemibarbus maculatus* larvae had a strong tolerance to hunger, having the ability to survive under starvation for up to 13 days.

## 5. Conclusions

The allometric growth pattern usually reflects the strategies through which fish larvae adapt to their living environment [2]. In this study, we found that the growth inflection point of *Hemibarbus maculatus* larvae was concentrated at TL = 10.93–16.53 mm, occurring between 14–20 DAH, according to the relationship between TL and DAH. *Hemibarbus maculatus* larvae preferentially developed their head, fins, and eyes, related to the functions of feeding, balance, and swimming, in order to cope with complex environments. The PNR of *Hemibarbus maculatus* larvae was 12–13 DAH, and the R value was 1.75, indicating that the larvae had strong hunger tolerance. The *Hemibarbus maculatus* larvae opened their mouth for feeding at 4 DAH. Therefore, when considering a water temperature of 22.66 ± 1.56 °C, 4–5 DAH is the best time to transfer to pond cultivation, and it should not be later than 12 DAH. Water quality and feeding management should be strengthened in the larvae of *Hemibarbus maculatus* before 20 DAH.

## Figures and Tables

**Figure 1 biology-13-00164-f001:**
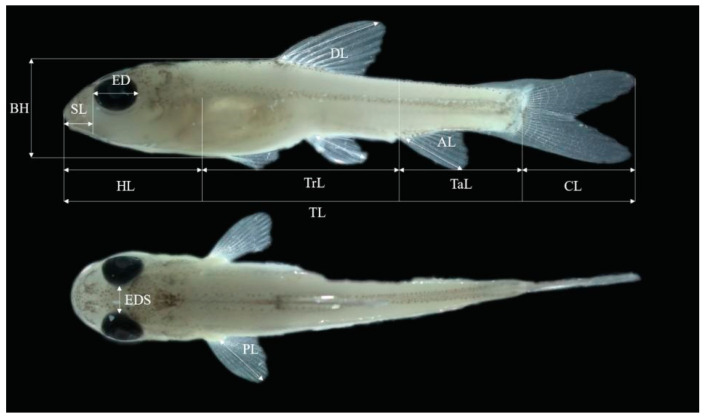
Schematic diagram of morphological and metric indicators in the *Hemibarbus maculatus* larvae. Total length (TL), head length (HL), snout length (SL), eye diameter (ED), eye distance (EDS), body height (BH), trunk length (TrL), pectoral fin length (PL), dorsal fin length (DL), caudal fin length (CL), anal fin length (AL), and tail length (TaL).

**Figure 2 biology-13-00164-f002:**
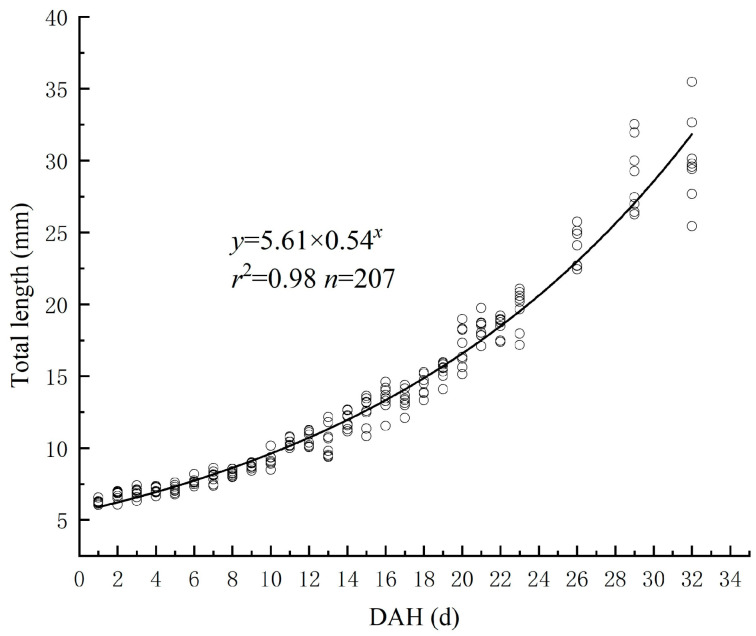
The relationship between TL and DAH.

**Figure 3 biology-13-00164-f003:**
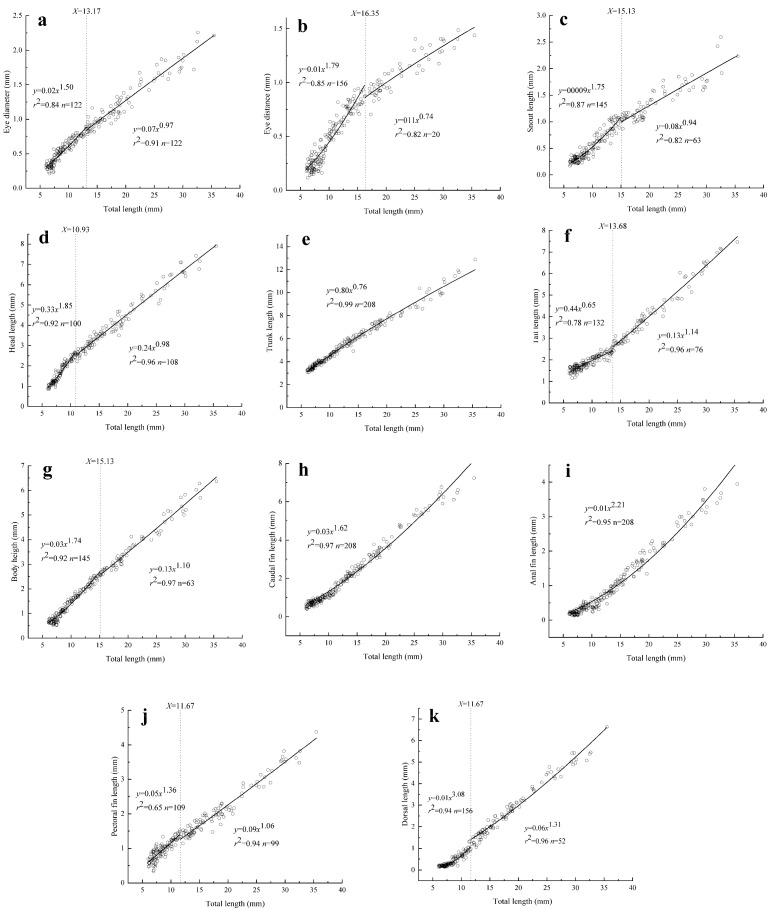
Allometric growth of various organs: (**a**–**k**) refer to allometric growth of eye diameter, eye distance, snout length, head length, trunk length, tail length, body length, caudal length, anal length, pectoral length, and dorsal length, respectively.

**Figure 4 biology-13-00164-f004:**
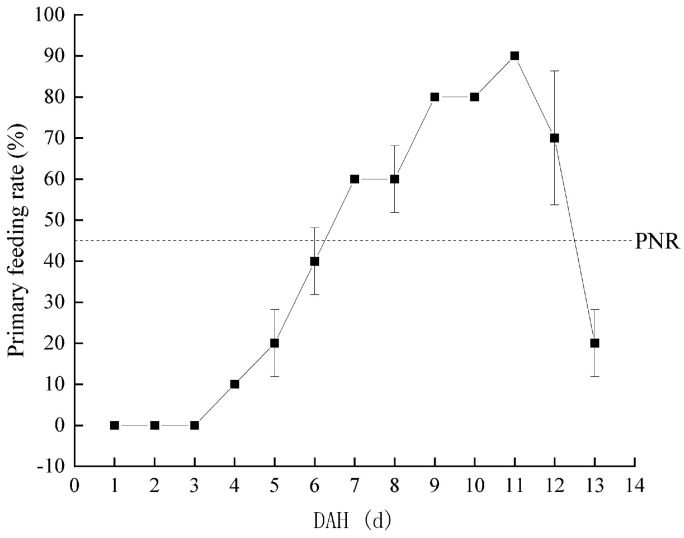
Change in primary feeding rate of *Hemibarbus maculatus* larvae.

**Figure 5 biology-13-00164-f005:**
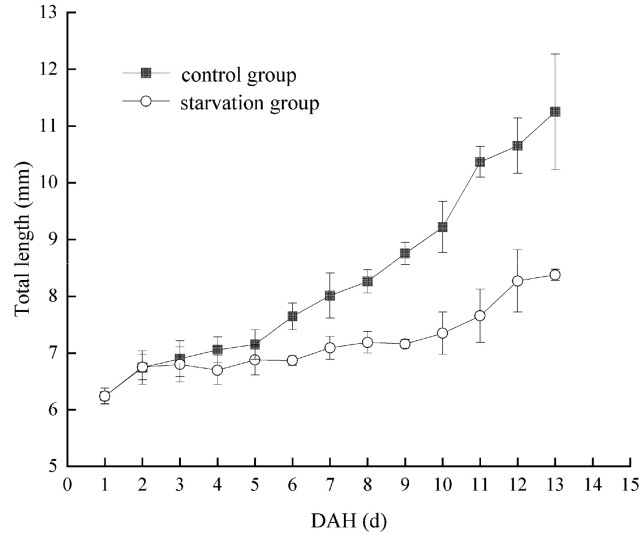
Comparison of change in total length of *Hemibarbus maculatus* larvae between control and starvation groups.

## Data Availability

Data are contained within the article.

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
