# Peer review of "Allometric Growth Pattern and Hunger Tolerance of Hemibarbus maculatus Bleeker Larvae"

_biology, 2024, doi:10.3390/biology13030164_

Round 1
Reviewer 1 Report (Previous Reviewer 2)
Comments and Suggestions for Authors
In abstract line 31:
"height were TL..." shoukl said: "height were at TL..."
You need to add the sample number of the analyses in the summary.
Add in Fig. 2 the “x” value the name should be change to “b” since “x” is TL and “b” the slope of the relationship
Authors need to review: Froese, R. 2006. Cube law, condition factor and weight–length relationships: history, meta-analysis and recommendations. Journal of Applied Ichthyology 22: 241–253.
Author Response
Dear reviewer:
We feel great thanks for your professional review work on our paper. As you are concerned, there are several problems that need to be addressed. According to your nice suggestions, we have made Moderate modification to our previous draft. Responses to the extract of referee opinions and suggestions for manuscript biology-2875723:
comment
"height were TL..." should said: "height were at TL..."
Response
Thank you for your careful check and reminder. We were really sorry for our careless mistake. As your suggestions, we have fixed the error using “Track-Changes” function of Word software.
comment
You need to add the sample number of the analyses in the summary.
Response
Thank you for your review. The control group and starvation group were set up in this study. 10 larvae were sampled from each group every day to study allometric growth pattern and starvation tolerance. We have added this sentence to the simple summary and abstract section.
comment
Add in Fig. 2 the “x” value the name should be change to “b” since “x” is TL and “b” the slope of the relationship
Response
Thanks for your comment. The figure2 was reflect The relationship between total length (TL) and day after hatching (DAH). Curve model fitting and parameter Estimation of relationship between TL and DAH of Hemibarbus maculatus larvae were performed using “Curve Estimation” in SPSS Statistics 23.0 software. We take the largest R2 as the best equation. The result of Curve model fitting and parameter Estimation as follow:
model |
R2 |
F |
df1 |
df2 |
sig |
Constant |
b1 |
b2 |
b3 |
Linear function model |
0.91 |
2103.52 |
1 |
206 |
0 |
2.708 |
0.756 |
||
Logarithmic curve model |
0.61 |
5518.84 |
1 |
206 |
0 |
-1.274 |
6.127 |
||
Conic function model |
0.971 |
3513.44 |
2 |
205 |
0 |
6.281 |
0.088 |
0.022 |
|
Cubic curve model |
0.972 |
2363.01 |
3 |
204 |
0 |
6.658 |
-0.44 |
0.032 |
0 |
Power function model |
0.775 |
712.187 |
1 |
206 |
0 |
3.86 |
0.475 |
||
Exponential curve model |
0.978 |
9180.16 |
1 |
206 |
0 |
5.61 |
0.54 |
  |
  |
Therefore, The best function was Y=5.61*0.54X R2=0.98.
comment
Authors need to review: Froese, R. 2006. Cube law, condition factor and weight–length relationships: history, meta-analysis and recommendations. Journal of Applied Ichthyology 22: 241–253.
Response
Thank you for your comment and recommended reference for us. We have read this paper carefully. This paper presents a historical review, a meta-analysis, and recommendations for users about weight-length relationships, condition factors and relative weight equations. But we did not measure the weight of Hemibarbus maculatus larvae. However, It is very helpful for us to carry out related research in the future.
Reviewer 2 Report (New Reviewer)
Comments and Suggestions for Authors
Reviewer comments to author
Manuscript Number: - 2875723
The manuscript entitled “Study on Allometric Growth and PNR (Point of No Reture) of 2 Hemibarbus maculatus Bleeker” is a nice piece of work. The author examines the growth pattern and hunger tolerance of Hemibarbus maculatus Bleeker larvae. It found that larvae open mouths for feeding 4 days after hatching and the yolk sac disappears at 11 days. The larvae have strong starvation tolerance and preferentially develop head, fins, and eyes for feeding, balance, and swimming.
First of all, the authors need to pay attention to the grammatical errors as well as the spelling errors and consistency throughout the text.
I suggest the quality of this manuscript is good for publication in the Biology Journal but it needs some minor grammatical corrections are required before publication. The following are comments for the authors.
Minor Comments
Many places with no space between numbers and units
Simple Summary
“In order to clarify allometric growth pattern and hunger tolerance of Hemibarbus maculatus Bleeker.” Change the phrase “To clarify the allometric growth pattern and hunger tolerance of Hemibarbus maculatus Bleeker”
Line number 21: (22.66±1.56)℃ space between numbers and units i.e., (22.66±1.56) ℃
Abstract
Authors are suggested to avoid the abbreviations in the abstract
Line number 31: space between numbers and units
Keyword
Keywords should be corrected in alphabetical order.
Introduction:
The ending of the introduction should lead to a recommendation of the study and please mention what is the environmental factors.
Material and methods
Line 107: To reduced variation should be to reduce variation
Please provide a reference for morphometrics
The body height (BH) measurement should be the maximum horizontal distance between the dorsal and ventral portions of the fish.
4) Trunk length (TrL) measurement should be the length from the anterior most point of the body to the mid-point of cloaca.
Results
The results part should be clearly elaborated. Statistical comparisons of the results are not clear. Need to be improved.
Discussion
Need to revise thoroughly for some of the phrases and spelling mistakes wherever applicable. surely Improved with highlights to the key findings points and their supporting appropriate references.
References
Please delete outdated references if possible 80 % of references should be from 2018 onwards.
Please crosscheck the style of references as per the journal format.
Comments on the Quality of English LanguageMinor editing of English language required
Author Response
Please see the attachment

Reviewer 3 Report (New Reviewer)
Comments and Suggestions for Authors
Present manuscript is well presented and satisfactory for publication after minor revisions as follows :
In regards to your manuscript title, is it possible than we add study on allometric growth pattern and hunger tolerance in Hemibarbus maculatus larvae in closed culture system,
1. Its a suggestion, but please add word larvae in title which reflect its importance to reader,
2. In abstract portion, instead of in this paper, please add word in this study or in the current study.
3. In regards to your refrences in text, please follow the journal pattern becz in some place you add year with numbering in text of methodology and discussions portion
4. In conclusion, add paragraph style instead of points, and also add the significance of your studies for future aquaculturist when used this species and would identify water temperature and hunger tolerance during allometric growth
5. In introduction portion add this word family before Cyprinidae
6. In methodology portion, please add the incubation water temperature.
Wish you best of luck
Regards
Reviewer
Author Response
Dear reviewer:
We feel great thanks for your professional review work on our paper. As you are concerned, there are several problems that need to be addressed. According to your nice suggestions, we have made Moderate modification to our previous draft. Responses to the extract of referee opinions and suggestions for manuscript biology-2875723.
comment
In regards to your manuscript title, is it possible than we add study on allometric growth pattern and hunger tolerance in Hemibarbus maculatus larvae in closed culture system. Its a suggestion, but please add word larvae in title which reflect its importance to reader.
Response
Thanks for your comment and suggestion. We have revised the manuscript title with “Track-Changes” function of Word software.
comment
In abstract portion, instead of in this paper, please add word in this study or in the current study.
Response
Thanks for your carefully review and suggestion. Based on your and other reviewers' comments, we have made appropriate modifications to these two sentences
comment
In regards to your refrences in text, please follow the journal pattern becz in some place you add year with numbering in text of methodology and discussions portion.
Response
Thank you for your careful reminder. We were really sorry for our careless mistake. We have fixed 6 errors in the article.
comment
In conclusion, add paragraph style instead of points, and also add the significance of your studies for future aquaculturist when used this species and would identify water temperature and hunger tolerance during allometric growth.
Response
Thanks for your comment and suggestion. We rewrote the conclusion with paragraph style. We also provide suggestion for aquaculturist abort cultivation of Hemibarbus maculatus larvae in the conclusion.
comment
In introduction portion add this word family before Cyprinidae
Response
Thank you for your careful reminder. We have add the “family” before Cyprinidae in introduction portion.
comment
In methodology portion, please add the incubation water temperature.
Response
Thanks for your comment and suggestion. The eggs was incubated at the water temperature ranged from 18.3-20.0 ℃. We have add the incubation water temperature in methodology portion.
Reviewer 4 Report (New Reviewer)
Comments and Suggestions for Authors
· It is better to do not use abbreviations in the title rather than using them in the keywords.
· It is better to use another choice of word instead of using “in order to” to start the sentence.
· The authors claimed that Hemibarbus maculatus has a contribution to the aquaculture industry, while there are some questions available that 1)how much is the production volume of this spices in the globe and the country? 2)is it profitable to culture this fish? Because I searched and found out that this fish mature at three years and what is the length/weight under a cultivation period?
· The authors claimed that measuring growth allometric and PNR are useful in fisheries resource conservation. How can this possible? In which ways? How can the results of this study help the resource conservation of H. maculatus? The authors can clarify these issues in the discussion section.
· Fig.1: Provide definition of all acronyms in the caption.
· What was the proximate composition of the commercial diet? How could the authors adjust daily feeding rate?
· Any ethical issue for fasting the fish until die?
· Did the authors use any homogeneity test in the data analysis? I reckon, some measured data need non-parametric tests in this study.
· L192: It is a big claim, since the appearance of the eye spot is the main indicator.
· The authors reached to this finding that the fish has a U growth pattern. Therefore, what is the application of this finding in the industry?
· L294: correct the statement by adding under optimal conditions and free from pathogens.
· L272: What criterion is considered for higher or lower resistance of fish to starvation period of time? Is there a specific numerical index for carps (Cyprinidae)? If not instead of this sentence, mention that H. maculatus have the ability to survive against hunger for up to ... days.
· Rephrase the conclusion section to be one comprehensive paragraph.
Author Response
Dear reviewer:
We feel great thanks for your professional review work on our paper. As you are concerned, there are several problems that need to be addressed. According to your nice suggestions, we have made Moderate modification to our previous draft. Responses to the extract of referee opinions and suggestions for manuscript biology-2875723.
comment
It is better to do not use abbreviations in the title rather than using them in the keywords.
Response
Thanks for your comment and suggestion. Based on your and other reviewers’ suggestion, we have made appropriate changes to the title.
comment
It is better to use another choice of word instead of using “in order to” to start the sentence.
Response
Thanks for your comment and suggestion. We have made appropriate corrections to this sentence in our resubmitted manuscript.
comment
The authors claimed that Hemibarbus maculatus has a contribution to the aquaculture industry, while there are some questions available that 1)how much is the production volume of this spices in the globe and the country? 2)is it profitable to culture this fish? Because I searched and found out that this fish mature at three years and what is the length/weight under a cultivation period?
Response
Thanks for your comment. Hemibarbus maculatus is a major economic fish widely distributed in various waters of China. Compared with four major Asia domestic carps, common carp and Siniperca chuatsi, the production volume of Hemibarbus maculatus is relatively small. Therefore, its production volume is not counted in China Fishery Statistical Yearbook. But its price is about ï¿¥35-45 yuan /kg, Higher than many farmed fish. In China, many provinces (such as Guangdong, Guangxi, Hunan, Hubei, Sichuan, etc.) are already breeding this fish.
Our team is setting industry standards about “Hemibarbus maculatus”. We have collected some biological data on Hemibarbus maculatus. We found that this fish mature at 1-2 years in Guangdong and Guangxi province, 2-3 years in Hunan and Sichuan province, 3-4 years in Liaoning province. We also measured the body length and weight of wild and farmed Hemibarbus maculatus populations of different ages. data are as follows:
Table 1 Body length and weight of wild population at different ages
age |
1+ |
2+ |
3+ |
4+ |
Body length (mm) |
71.0-185.0 |
126.0-237.0 |
169.9-310.0 |
204.0-330.0 |
Weight (g) |
6.0-255.9 |
69.7-255.9 |
258.1-485.0 |
284.1-580.0 |
Table 2 Body length and weight of farmed population at different ages
age |
0+ |
1+ |
2+ |
3+ |
Body length (mm) |
75.8-109.4 |
130.0-192.0 |
193.0-266.0 |
292.0-332.0 |
Weight (g) |
5.4-17.0 |
24.2-76.2 |
123.5-314.7 |
288.2-627.3 |
Due to the differences in people's eating habits in different parts of China, this fish can be marketed in excess of 50g. In other words, more than 1+ age can be marketed.
comment
The authors claimed that measuring growth allometric and PNR are useful in fisheries resource conservation. How can this possible? In which ways? How can the results of this study help the resource conservation of H. maculatus? The authors can clarify these issues in the discussion section.
Response
Thanks for your review. We have clarified these issues in the discussion and introduction section.
comment
Fig.1: Provide definition of all acronyms in the caption.
Response
Thanks for your comment and suggestion. We have added the definition of all acronyms in the caption of figure1 in our resubmitted manuscript.
comment
What was the proximate composition of the commercial diet? How could the authors adjust daily feeding rate?
Response
Thanks for your review. The commercial diet used in this study was the compound feed produced by Zhejiang Jinjia Aquatic Feed Co., LTD (Crude protein ≥ 43.0%, Crude lipid ≥ 5.0%, Moisture ≤ 10.0%, Ash ≤ 18.0%). Its raw materials: white fish meal, α-starch, brewer's yeast, vitamins, minerals, probiotics and amino acids and so on. The experiment was fed twice a day and overfed each time.
comment
Any ethical issue for fasting the fish until die?
Response
There was no ethical issue in this study. It is a method to study the PNR and hunger tolerance. Many scholars have used this method to study the starvation tolerance of fish larvae. In this study, the animal research was approved by the Animal Care Committee of Hunan Fisheries Science Institute, Changsha, China (Approval Code: No. HFSI2020013). All the experimental phases were strictly controlled.
comment
Did the authors use any homogeneity test in the data analysis? I reckon, some measured data need non-parametric tests in this study.
Response
Thanks for your comment and reminder. Before the t test the homogeneity-of-variance was conducted, and when the Sig value was greater than 0.05, the variance could be considered homogeneous. We have added this sentence in the “2.3 data analysis”
comment
L192: It is a big claim, since the appearance of the eye spot is the main indicator.
Response
Thanks for your comment. In fish, the eye spot has already appeared in the fertilized egg stage. Early research suggested that the head's priority development could provide enough space for the eyes, brain, and gills, which are related to vision, neural information processing, and breathing (Xie et al 2023; Wang et al., 2016)
Xie M., Wu H., Song R., Xiang J., Zhou J., Li Hong., Zeng G Q., Li S. M., Xiang J G. 2023. Allometric growth pattern and point of no return for starvation of Furong crucian carp (Carassius auratus Furong carp ♀×Cyprinus carpio red crucian carp ♂) Journal of Southern Agriculture. 2023,54(4):1253-1262. Doi:10.3969/j.issn.2095-1191.2023.04.030.
Wang Y F, Xiao Z Z, Liu Q H, Zhai J M, Pang Z F, Ma W H, Ma D Y, Xu S H, Xiao Y S, Li J. Allometric growth pattern during early ontogeny of spotted knifejaw (Oplegnathus punctatus). Marine Sciences. 2016, 40(5): 43-48. Doi: 10.11759//hykx20141216005
comment
The authors reached to this finding that the fish has a U growth pattern. Therefore, what is the application of this finding in the industry?
Response
Thanks for your comment. Knowledge about the allometric growth patterns during the early life history stages of a species provides essential information on the adaptation strategies of fish populations. For example, pelagic larvae usually experience rapid development in their anterior and posterior regions to generate an elongated body, which improves swimming capabilities and feeding success (Snik et al. 1997). In contrast, benthic larvae usually have rapid development of their head and trunk width, which creates a short and fattened morphotype suitable for slow-moving bottom dwellers (Mello et al. 2015). In our study The development of body parts of Hemibarbus maculatus larvae roughly conforms to the "U" growth pattern. It may be suggest that swimming ability was not particularly important for larvae of Hemibarbus maculatus. It was similar with benthic and Siniperca chuatsi larvae (Mello et al. 2015, Song et al. 2018)
Mello GCG, Santos JE, Guimarães-Cruz RJ, Godinho AL, Godinho HP. Allometric larval growth of the bottom-dwelling catfish Lophiosilurus alexandri Steindachner, 1876 (Siluriformes: Pseudopimelodidae). Neotrop Ichthyol 2015, 13:677–684
Van Sink. G M J, Van den boogaart J G M, Osse J W M. Larval growth patterns in Cyprinus carpio and Clarias gariepinus with attention to the finfold. Journal of Fish Biology, 1997, 50: 1339-1352. Doi: 10.1111/j.1095-8649.1997.tb01657.x
Song, Y Q, Cheng F, Zhao S S, Xie S G. Ontogenetic development and otolith microstructure in the larval and juvenile stages of mandarin fish Siniperca chuatsi. Ichthyological Reseach, 2019, 66: 57-66. Doi: 10.1007/s10228-018-0648-1
comment
L272: What criterion is considered for higher or lower resistance of fish to starvation period of time? Is there a specific numerical index for carps (Cyprinidae)? If not instead of this sentence, mention that H. maculatus have the ability to survive against hunger for up to ... days.
Response
Thanks for your review and suggestion. Shan et al. (2008) proposed that the ratio (R) of DAH of mixed trophic period to endotrophic period was used as the criterion for discrimination . If R>1, it indicates that larvae are less susceptible to starvation stress. Otherwise, they are vulnerable to starvation stress. In this study, R=1.75>1, indicating that Hemibarbus maculatus larvae are less susceptible to starvation stress. At the end of the discussion section, we also added this sentence “Hemibarbus maculatus have the ability to survive against hunger for up to 13 days.”
comment
Rephrase the conclusion section to be one comprehensive paragraph.
Response
Thanks for your comment and suggestion. As your suggestion, we rewrote the conclusion with paragraph style.
Round 2
Reviewer 4 Report (New Reviewer)
Comments and Suggestions for Authors
No further comments and suggestions.
This manuscript is a resubmission of an earlier submission. The following is a list of the peer review reports and author responses from that submission.
Round 1
Reviewer 1 Report
Comments and Suggestions for Authors
General comments:
The manuscript is original. According to my opinion, this paper represents a valuable contribution to the knowledge regarding hunger tolerance and trends in growth in the species Hemibarbus maculatus Bleeker. All parts of the paper are enough, described and illustrated properly but not fully. The introduction section should be rewritten to contain more information regarding the background of the topic, especially the biology and ecology of the investigated species. Moreover, the main goal of the study is not clearly emphasized. Since the species is defined as economically important, authors must highlight in conclusion section the importance of the results obtained as well their implications. Also, there are too many technical mistakes in the text.
Minor comments:
Page1 Line 12-13 and 23-24. The first sentence looks incomplete. Please rewrite.
Page1 Line 16. Abbreviations mentioned the first time in the text (i.e. PNR) must be defined
Page 2 Line 60-67. These sentences are not in the correct place in the text. It seems this paragraph belongs to the Material and Methods section.
Page 2 Line 63. The Latin name is not written correctly
Page 3 Line 101. The heading should start with a capital letter.
Page 3 Line 124. Prior to any statistical testing normal distribution should be analyzed. This further defines which statistical test will be used.
Page 3 Line 131. R2 instead of R2
Page 4. Line 133 Figure 2 instead of figure 2 (this refers for all text)
Page 4. Line 139 significantly different instead of Significantly greater
Page 4. Line 150 New sentence starts with a capital letter
Page 7 Line 194 Use either Latin or common names of fish
Plage 7 Line 213 The same comment
The written English is extremely poor. Consequently, some parts of the manuscript are hardly understandable. A native speaker must correct the language.
Author Response
Reviewer 1*
*Comment*
1) The introduction section should be rewritten to contain more
information regarding the background of the topic, especially the
biology and ecology of the investigated species.Moreover, the main goal
of the study is not clearly emphasized. Since the species is defined as
economically important, authors must highlight in conclusion section the
importance of the results obtained as well their implications. Also,
there are too many technical mistakes in the text.
*Response*
Thank you for your patient comments. We have added a description of the
biology, nutritional value and ecology of /Hemibarbus maculatus/in the
introduction. We are sorry for our carelessness for this technical
mistakes in our paper. If you think our manuscript should undergo
extensive English revisions, we were willing to use the English language
editing services of MDPI.
*Commet*
2)Line 12-13 and 23-24. The first sentence looks incomplete. Please rewrite.
*Reponse*
Thanks for your careful checks. This sentence is mainly used to describe
the period of days of endogenous nutrition and exogenous nutrition. In
order to more objectively reflect the hunger tolerance of different
species of larvae, Shan et al. (2008) proposed that the ratio (R) of DAH
of mixed trophic period to endotrophic period was used as the criterion
for discrimination. If R>1, it indicates that larvae are less
susceptible to starvation stress. Otherwise, they are vulnerable to
starvation stress.
References : Shan X J, Dou S Z. Effect of delayed first feeding on
growth, survival and biochemical composition of Croaker Michthys Miiuy
Larvae. Oceanologia Et Limnologia Sinica. 2008, 39(1):14-23.
*Comment*
3) Line 16. Abbreviations mentioned the first time in the text (i.e.
PNR) must be defined
Line 63. The Latin name is not written correctly
Line 101. The heading should start with a capital letter.
Line 124. Prior to any statistical testing normal distribution should be
analyzed. This further defines which statistical test will be used.
Line 131. R2 instead of R2
Line 133 Figure 2 instead of figure 2 (this refers for all text)
Line 139 significantly different instead of Significantly greater
Line 150 New sentence starts with a capital letter
Line 194 Use either Latin or common names of fish
Line 213 The same comment
*Respons*
Thanks for your careful checks. We are sorry for our carelessness. Based
on your comments, We have fixed it. Unfortunately, we thought the
editorial department was not interested in this article, and we did not
use “Track-Changes” function of Word software.
Reviewer 2 Report
Comments and Suggestions for Authors
There is an error in the equation of Figure 2, it should be: y = 5.61*x0.54
Author Response
*Reviewer 2*
*Comment*
1) There is an error in the equation of Figure 2, it should be: y =5.61*x^0.54
*Respons*
Thanks for your careful checks. We are sorry for our carelessness. Based on your comments, We have fixed it in our paper.
Reviewer 3 Report
Comments and Suggestions for Authors
1. “Simple Summary” and “Conclusions” are highly similar, different expressions of two parts should be adopted.
2, “Introduction” is too simple and content should be added.
3. L.264: "It indicated that Hemibarbus maculatus larvae had strong tolerance to hunger." What are the criteria for judging the strength of the tolerance to hunger?
Author Response
Reviewer 3*
*Comment*
1) “Introduction” is too simple and content should be added.
*Reponse*
Thank you for your patient comments. We have added a description of the
biology, nutritional value and ecology of /Hemibarbus maculatus/in the
introduction.
2) L.264: "It indicated that /Hemibarbus maculatus/ larvae had strong
tolerance to hunger." What are the criteria for judging the strength of
the tolerance to hunger?
*Reponse*
In order to more objectively reflect the hunger tolerance of different
species of larvae, Shan et al. (2008) proposed that the ratio (R) of DAH
of mixed trophic period to endotrophic period was used as the criterion
for discrimination. If R>1, it indicates that larvae are less
susceptible to starvation stress. Otherwise, they are vulnerable to
starvation stress. In this study, R=1.75>1, indicating that /Hemibarbus
maculatus/ larvae are less susceptible to starvation stress. In
addition, the PNR of /Hemibarbus maculatus/was located between 12 and 13
DAH. This is similar to many fish species that have a strong strength of
the tolerance to hunger. Therefor, we think that /Hemibarbus maculatus/
larvae had strong tolerance to hunger.
References : Shan X J, Dou S Z. Effect of delayed first feeding on
growth, survival and biochemical composition of Croaker Michthys Miiuy
Larvae. Oceanologia Et Limnologia Sinica. 2008, 39(1):14-23.
Reviewer 4 Report
Comments and Suggestions for Authors
This manuscript is difficult to understand. I am not going to comment point by point. There is even abbreviation in the title without explanation, i.e., PNR.
Moreover, this study was too descriptive, lacking a rigid experimental design.
Comments on the Quality of English LanguageIt is difficult to understand the research content.
Author Response
*Reviewer 4*
*Comment*
1) This manuscript is difficult to understand. I am not going to comment point by point. There is even abbreviation in the title without explanation, i.e., PNR.
*Reponse*
Thank you for comment. We are sorry about our English writing. If you
think our manuscript should undergo extensive English revisions, we were
willing to use the English language editing services of MDPI. We have
added the explanation of the PNR in the title.